# Evaluation of the test accuracy of a SARS-CoV-2 rapid antigen test in symptomatic community dwelling individuals in the Netherlands

**Nathalie Van der Moeren** [1][*], **Vivian F. Zwart** [1], **Esther B. Lodder** [2], **Wouter Van den Bijllaardt** [1], **Harald R. J. M. Van Esch** [1], **Joep J. J. M. Stohr** [1], **Joost Pot** [2], **Ineke Welschen** [2], **Petra M. F. Van Mechelen** [2], **Suzan D. Pas** [1], **Jan A. J. W. Kluytmans** [1,3]

1 Microvida Laboratory for Medical Microbiology, Amphia Hospital, Breda, The Netherlands, 2 GGD West-Brabant, Breda, The Netherlands, 3 Julius Center for Health Sciences and Primary Care, University Medical Center Utrecht, Utrecht University, Utrecht, The Netherlands

☯ These authors contributed equally to this work.

* n.vdmoeren@gmail.com

**Data Availability Statement:** All relevant data will be provided within a Supporting Information file.

## Abstract

### Background

SARS-CoV-2 real-time reverse transcriptase polymerase chain reaction (qRT-PCR) is well suited for the diagnosis of clinically ill patients requiring treatment. Application for community testing of symptomatic individuals for disease control purposes however raises challenges. SARS-CoV-2 rapid antigen tests might offer an alternative, but quality evidence on their performance is limited.

### Methods

We conducted an evaluation of the test accuracy of the 'BD Veritor System for Rapid Detection of SARS-CoV-2' (VRD) compared to qRT-PCR on combined nose/throat swabs obtained from symptomatic individuals at Municipal Health Service (MHS) COVID-19 test centers in the Netherlands. In part one of the study, with the primary objective to evaluate test sensitivity and specificity, all adults presenting at one MHS test center were eligible for inclusion. In part two, with the objective to evaluate test sensitivity stratified by Ct (cycle threshold)-value and time since symptom onset, adults who had a positive qRT-PCR obtained at a MHS test center were eligible.

### Findings

In part one (n = 352) SARS-CoV-2 prevalence was 4.8%, overall specificity 100% (95%CI: 98·9%-100%) and sensitivity 94·1% (95%CI: 71·1%-100%). In part two (n = 123) the sensitivity was 78·9% (95%CI: 70·6%-85·7%) overall, 89·4% (95% CI: 79·4%-95·6%) for specimen obtained within seven days after symptom onset and 93% (95% CI: 86%-97.1%) for specimen with a Ct-value below 30.

**Funding:** The authors received no specific funding for this work. The antigen tests for the study were provided by the Dutch Ministry of Health, Welfare and Sport (VWS). The VWS was not actively involved in the design, execution, analysis or result interpretation of the study, nor in the writing of the manuscript or the submission for publication.

**Competing interests:** The authors have declared that no competing interests exist.

## Interpretation

The VRD is a promising diagnostic for COVID-19 testing of symptomatic community-dwelling individuals within seven days after symptom onset in context of disease control. Further research on practical applicability and the optimal position within the testing landscape is needed.

## Background

Accurate and sustainable test strategies are essential for the control of SARS-CoV-2 [1]. The current test used to establish an acute SARS-CoV-2 infection in The Netherlands is real-time reverse transcriptase polymerase chain reaction (qRT-PCR). This test is highly sensitive and specific and therefore well suited for the diagnosis of clinically ill patients. However, application of the test for large-scale community testing of symptomatic individuals for disease control purposes raises substantial challenges. qRT-PCR can only be performed in specialised laboratories, has a relatively long turnaround time (TAT) and depends on the availability of scarce extraction and PCR reagents and disposables. The massive qRT-PCR demand created by community screening, greatly burdens microbiological laboratories and puts routine clinical diagnostic care at risk. Furthermore, logistic and administrative challenges lead to substantial delays in testing and reporting of the results. Rapid testing and reporting are however key in the control of SARS-CoV-2 community spread [2]. COVID-19 community screening requires a low-cost diagnostic test with a short TAT which can be performed close to the community. Lateral flow assay (LFA) SARS-CoV-2 antigen tests can be performed at point of care, give results within 15–30 minutes and are relatively inexpensive to produce [3, 4]. Numerous SARS-CoV-2 LFA are available, but quality data on their performance is limited. The available studies are often based on remnant laboratory samples and contain little information on the clinical setting or disease stage. The current literature is insufficient to determine whether SARS-CoV-2 rapid antigen test can be useful in clinical practice and prospective evaluation of the antigen tests in clinically relevant settings is needed [5].

Hence, this evaluation of the test accuracy of the 'BD Veritor System for Rapid Detection of SARS-CoV-2' (VRD) when performed on combined nose/throat swabs obtained from symptomatic individuals at two COVID-19 test centers of the Dutch Municipal Health Service (MHS).

## Methods

### Objectives

The primary study objective of the prospective performance evaluation in part one of the study was to determine the specificity and sensitivity on clinical samples of the VRD compared to qRT-PCR. Secondary objective was to evaluate the concordance between visual interpretation of VRD test results and analysis using the reading device provided by the manufacturer, the BD Veritor Analyzer (VA).

The primary objective of part two of the study was to determine the sensitivity for different Ct-value groups (Ct <20, Ct 20–25, Ct 25–30 and Ct ≥30) and different intervals since symptom onset (< 7 days, ≥ 7 days).

## Setting

COVID-19 testing of non-hospitalized symptomatic patients in the Netherlands is coordinated by the MHS. Individuals can–provided they state to have COVID-19 like symptoms (rhinitis, cough, elevated temperature (not further specified), shortness of breath or sudden loss of sense of taste or smell)—make an appointment at a regional MHS test center. These criteria remained unchanged during the study period. A single swab is used to collect the specimen from throat and nose and is sent to the microbiological laboratory for qRT-PCR. Swabs are obtained by specifically trained MHS employees, who do not always have a medical background. Individuals with a positive qRT-PCR result are informed by a MHS employee and approached with a questionnaire for the purpose of source- and contact- tracing.

The study was conducted from September 26th to October 7th 2020 in the region West-Brabant, the Netherlands. The local MHS had three operational test centers during the study, conducting 1200 SARS-CoV-2 qRT-PCRs daily. Because of logistic reasons (travel distance to the laboratory), part one of the study was performed at one MHS center (Breda). As a portion of the samples of one of the three test centers were sent to an external laboratory, samples from the two centers (Breda and Roosendaal) were considered for part two of the study. In the third week of September 2020 5–6% of individuals tested at a West Brabant MHS test center had a positive qRT-PCR (data on file).

## BD veritor system for rapid detection of SARS-CoV-2 (VRD)

The VRD is a chromatographic lateral flow immunoassay for the detection of SARS-CoV-2 nucleocapsid antigens in respiratory specimen. The manufacturer reports a test specificity of 100% (95%CI: 98%-100%) and a sensitivity of 84% (95%CI: 67%-93%) compared to qRT-PCR as a reference standard during the first 5 days after disease onset. The test was validated by the manufacturer for use on superficial nasal specimen. The manual prescribes interpretation of the results after 15 minutes with a reading device provided by the manufacturer (VA) [3]. Nevertheless, a test and control line can be seen with the naked eye.

## Real-time reverse transcriptase PCR

Two CE-IVD labelled qPCR platforms were used according to manufacturer's protocols. Firstly, the Cobas 6800 (Roche) platform using Cobas® SARS-CoV-2–192 PCR assay (Roche diagnostics), detecting RdRp and E-genes. Secondly, the m2000SP and m2000RT platform (Abbott) was used in combination with the Abbott mSample prep. System kit and the Abbott Real Time SARS-CoV-2 Amplification Reagent kit, detecting both E-gene and N-gene. Swabs for qRT-PCR were stored in a 1:1 lysis buffer: virus transport medium.

## Patient recruitment

In part one of the study all adults ($\geq$18 years) presenting at the MHS test center Breda for a COVID- 19 test between September 28 and 30 2020 were invited to participate. Individuals who were able and willing to give verbal informed consent were included.

In part two of the study, adults ($\geq$18 years) who had been tested at one of the two included MHS test centers between September 26 and October 6 and had a positive qRT-PCR were approached by a MHS employee and invited to participate. Individuals who were able and willing to give verbal informed consent and who confirmed to be or have been symptomatic and who had a positive qRT-PCR at the moment of the home visit (see below 'study procedure') were included.

## Ethics

The planning, conduct and reporting of the study was in line with the Declaration of Helsinki, as revised in 2013. The study was registered at the Netherlands Trial Register (identification number NL9018).

In part one of the study, individuals were informed about the study through local media, by MHS communication channels (full participant information letter on the website) and by information signs at the participating test centers. Verbal informed consent was obtained separately by two independent MHS employees. No written informed consent was obtained as this would have compromised the strictly needed high flow of individuals being tested at the test centers (3 minutes per client). In part two of the study, potential participants were informed about the study and asked for verbal informed consent a first time by telephone. Verbal informed consent was obtained by a different MHS employee a second time during a home visit before obtainment of the study samples. No written informed consents were obtained as handling of documents obtained from confirmed infectious participants was considered a potential safety hazard.

The study protocol was submitted at the medical ethical board ´Medical research Ethics Committees United´ (MEC-U) and was granted an exemption of the Dutch medical scientific research act (WMO).

## Study procedure

In part one of the study, one swab was used to obtain a specimen from the throat and nasal cavity up to the nasal bridge for routine qRT-PCR in accordance with the Dutch national COVID-19 test protocol. In addition to and directly following this first swab, the same MHS employee obtained an additional swab to acquire a specimen from the throat and the superficial nasal cavities (bilateral, 2·5 cm proximal from the nostril) for VRD. The swabs for VRD were immediately deposited in dry in sterile test tubes and stored and transported on dry ice until processing at the laboratory. The VRD were performed by trained laboratory technicians within 6 hours after obtainment of the sample. Samples were left 15 minutes at room temperature before analysis in accordance with the manufacturer's operating procedure. Test results were read visually after 15 minutes and thereafter with the VA. No clinical information or information on qRT-PCR results were available to the technicians performing the VRD. Information on the first day of illness was subtracted later on from the MHS files intended for source and contact tracing.

In part two of the study, participants were visited at home by MHS employees within 72 hours after their initial positive qRT-PCR at the MHS test center. Analogous to the procedure in part one of the study, specimens for both qRT-PCR and VRD were obtained, stored and analyzed. Only the results of visual interpretation were withheld. In addition, participants were asked what the day of symptom onset was and whether they still had symptoms at the time of the home visit.

## Sample size

The sample size calculation was based on an expected sensitivity of 80% in accordance with the performance data reported by the manufacturer [3]. Based on a margin of error of 7%, type I error of 5% and power of 80%, we aimed to include 125 qRT-PCR positive samples.

## Analysis

The primary outcome of part one of the study was the VRD sensitivity and specificity on clinical samples compared to qRT-PCR, based on interpretation of the results with the naked eye

and with the VA. Furthermore, overall positive predictive value (PPV) and negative predictive value (NPV) were calculated for a population prevalence of 10 and 20% using Medcalc version 19.6.4. For part two of the study the primary outcome was VRD sensitivity compared to qRT-PCR stratified by Ct-value category (Ct<20, Ct20-25, Ct25-30 and Ct≥30) and time since symptom onset (< 7 days, ≥ 7 days). The 7-day cut-off was based on the results of Bullard et al. showing no viral growth in Vero cells in samples obtained over 8 days after symptom onset [5]. Differences between groups were compared using chi-squared tests with n-1 correction for categorical variables. Clopper–Pearson Exact confidence intervals were calculated for sensitivity and specificity. All data were analyzed using Excel, Medcalc version 19.6.4. and SPSS version 24.

## Results

In part one of the study 354 individuals, men and women aged 18 years and above, who presented at the test center were initially included. A diagram of the flow of participants is shown in Fig 1. Two (0·6%) specimens with a negative VRD result were excluded because qRT-PCR could not be recovered (error in sample number registration). 17 samples had detectable SARS-CoV-2 RNA, resulting in a prevalence of 4.8 per 100 participants. Amongst the 17 qRT-PCR positive specimen 12 (70·6%) were obtained within seven days after disease onset, one (5·9%) was obtained later and for four specimens (23·5%) the time since symptom onset could not be determined. One qRT-PCR negative specimen rendered an uninterpretable and invalid VRD result by respectively visual interpretation and interpretation with the analyzer. VRD was positive for 16 specimens based on visual interpretation and for 18 specimens based

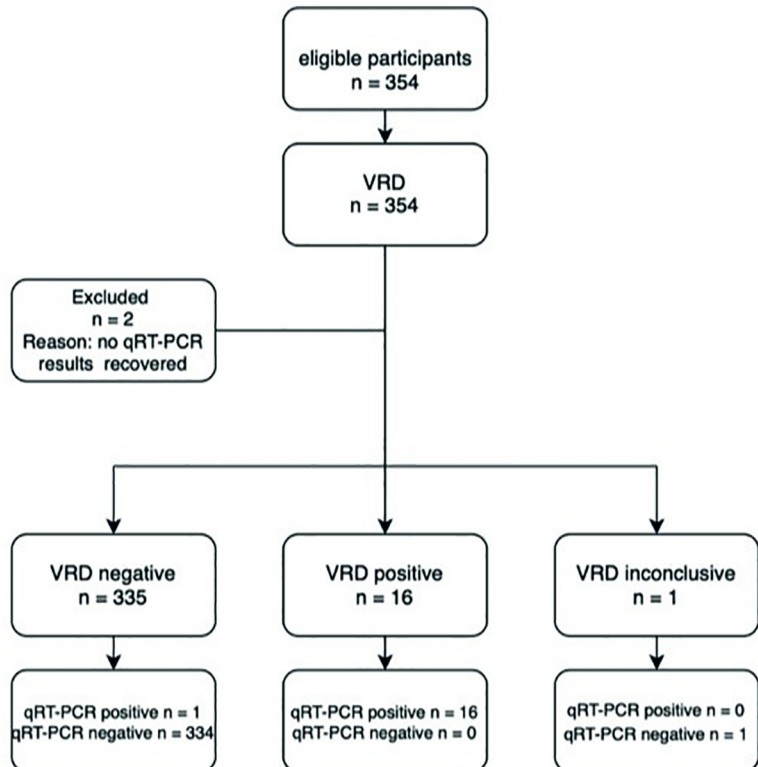

**Fig 1. Diagram for the flow of participants for part 1 of the study (prospective cohort).**

**Table 1. VRD performance compared to qRT-PCR in study part one.**

|  | Visual interpretation | Interpretation with analyzer |
|---|---|---|
| Total (n) | 352 | 352 |
| Invalid | 1 | 1 |
| True positive (n) | 16 | 16 |
| False positive(n) | 0 | 2 |
| True negative (n) | 334 | 332 |
| False negative (n) | 1 | 1 |
| Sensitivity (%) [95% CI] | 94·1% [71·1%-100%] | 94·1% [71·1%-100%] |
| Specificity (%) [95% CI] | 100% [98·9%-100%] | 99·4% [97·9%-100%] |

on interpretation with the VA. The two samples which were positive with the analyzer and negative by visual reading had a negative qRT-PCR. All 16 samples which were positive based on visual interpretation were qRT-PCR positive. Specificity was 100% (95%CI: 98·9%-100%) based on visual interpretation and 99·4% (95%CI: 97·9%-100%) based on interpretation with the analyzer, the sensitivity was 94·1% (95%CI: 71·1%-100%) (Table 1). The single qRT-PCR positive sample that was tested negative with VRD had a Ct-value of 32·7 and unknown time since symptom onset.

For the prevalence of 4.8% in the study cohort, the positive predictive value (PPV), based on visual interpretation of the test results, was 100% and negative predictive value (NPV) was 99.7% (95% CI 98.1%-99.7%). always 100% as specificity in this case was 100%. The NPV was above 98% for a population prevalence up to 20%, PPV was always 100% as specificity was 100% (Table 2).

In part two of the study, initially 132 participants were eligible for inclusion. Three individuals were excluded as they stated not to have been symptomatic at any point in time. One of them was tested qRT-PCR and VRD positive, one qRT-PCR positive and VRD negative and one was tested negative in both. Furthermore, six (4·5%) symptomatic individuals had a negative qRT-PCR at time of the home visit, all of them had a negative VRD (Fig 2). The ages of the 123 finally included individuals varied from 18 to 83 years (Mean = 44, SD = 16). All but one Ct-value were obtained by the Roche 6800 qRT-PCR. The one exception which was tested on the Abbott platform and had a Ct-value below 20. The sensitivity of the VRD in symptomatic individuals was 78·9% (95%CI: 70·6%-85·7%). When stratified by Ct-value category, sensitivity was found to be higher in the lower Ct-value categories (higher viral loads) compared to the highest Ct-value category (Ct<20 100% (95%CI: 83·2%-100%) (p<0.001), Ct20-25 93·3% (95%CI: 81·7%-98·6%) (p<0.001), Ct25-30 88·2% (95%CI: 72·6%-96·7%) (p<0.001), Ct>30 20·8% (95%CI: 7·1%-42·2%)). When subdivided in time since symptom onset shorter than seven days or seven days or more, clinical sensitivity was higher for those specimens obtained within seven days after symptom onset overall and for every Ct-value category (p = 0.002) (Table 3).

**Table 2. Negative predictive values (NPV) and positive predictive values (PPV) based on visual interpretation of VRD results for different population prevalence.**

|  | Population prevalence | | |
|---|---|---|---|
|  | 4.8% | 10% | 20% |
| NPV (%) [95% CI] | 99.7% [98.1%-99.7%] | 99.4% [95.8%-99.9%] | 98.6% [91.0%-99.8%] |
| PPV (%) | 100% | 100% | 100% |

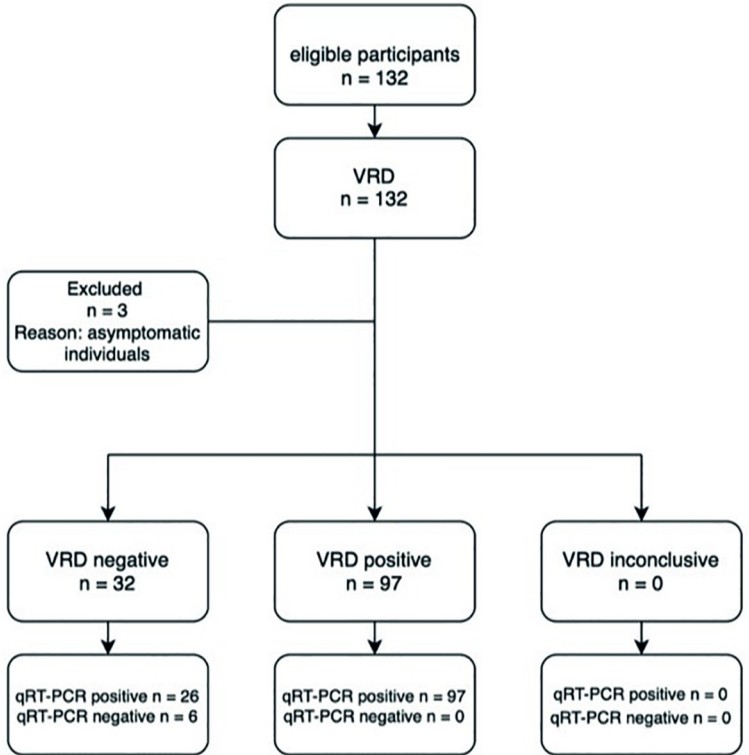

**Fig 2. Diagram for the flow of participants for part 2 of the study (qRT-PCR positive participants only).**

## Discussion

We found an overall clinical specificity of 100% (95% CI: 98·9%-100%) and sensitivity of 94·1% (71·1%-100%) of the VRD, when results were interpreted visually, compared to qRT-PCR based on the prospective cohort in part one of the study. For the cohort in part two of the study, sensitivity was higher for lower Ct-value categories (p< 0.001) and for specimen obtained within the first days after disease onset (p = 0.002). For specimen obtained within

**Table 3. Test results of 123 qRT-PCR positive specimen of symptomatic individuals from study part two.**

| Days since symptom onset | Ct-value category | qRT-PCR + samples (n) | VRD + (n) | VRD—(n) | Sensitivity (%) [95% CI] |
|---|---|---|---|---|---|
| < 7 days | Ct < 20 | 17 | 17 | 0 | 100% [80·1%-100%] |
| | Ct 20–25 | 29 | 29 | 0 | 100% [88·1%-100%] |
| | Ct 25–30 | 12 | 11 | 1 | 91·7% [61·5%-99·8%] |
| | Ct ≥ 30 | 8 | 2 | 6 | 25·0% [3·2%-65·1%] |
| | Overall | 66 | 59 | 7 | 89·4% [79·4%-95·6%] |
| | **CT < 30** | 58 | 57 | 1 | 98·3% [90·8%-100%] |
| ≥ 7 days | Ct < 20 | 3 | 3 | 0 | 100% [29·2%-100%] |
| | Ct 20–25 | 16 | 13 | 3 | 81·3% [54·4%-96·0%] |
| | Ct 25–30 | 22 | 19 | 3 | 86·4% [65·1%-97·1%] |
| | Ct ≥ 30 | 16 | 3 | 13 | 18·8% [4·1%-45·7%] |
| | Overall | 57 | 38 | 19 | 66·7% [52·9%-78·6%] |
| | **CT < 30** | 41 | 35 | 6 | 85·4% [70·8%-94·4%] |

Ct-value: cycle threshold value, qRT-PCR: real-time reverse transcriptase polymerase chain reaction, VRD: 'BD Veritor System for Rapid Detection of SARS-CoV-2'.

seven days after symptom onset the sensitivity was 89·4% (95%CI 79·4%-95·6%) overall and 98·6% (98·3% (95%CI 90·8%-100%) for samples with qRT-PCR Ct-value beneath 30.

To our knowledge no independent validation reports on the VRD have been published to date. Although numerous SARS-CoV-2 antigen tests are available on the market, quality data on test performance is limited. A review identified 5 performance evaluation studies evaluating a total of 8 SARS-CoV-2 antigen tests. The reported average specificity of 99·5% (95% CI: 98·1% to 99·9%) was in line with the results of our study. Sensitivity varied strongly across studies (from 0% to 94%) with an average of 56·2% (95% CI: 29·5% - 79·8%). The included studies were performed on remnant specimen stored in virus transport medium and often contained little information on days since disease onset and the clinical setting they were obtained in, all possibly explaining the discrepancy with the observed clinical sensitivity of the VRD in this study [5]. Preliminary results of two performance evaluation studies of the Panbio Antigen test (Abbott) with to this study similar protocols performed on a total of 1397 samples were largely in line with the results observed here: an overall specificity of 100% and sensitivity of 73·2% (Utrecht) and 81·8% (Aruba). Similar to the VRD, the Panbio Antigen test was reported to perform better for lower Ct-value categories [6].

The prospective design of part one of the study and the obtainment of samples in the target setting of potential use are great assets of this study.

As waiting times to make an appointment at a COVID-19 test center were long during the study period due to the great demand, no specimen collected within two days after disease onset could be included. Although we expect this group to have high viral loads, we cannot ascertain this assumption. The lack of data on this early window is a limitation of the study.

As samples for part two of the study were gathered during a house visit 24–48 hours after the initial positive test, participants in this cohort were likely to be further in the disease process on average compared to the population presenting at the MHS health centers.

When stratifying the results of part two of the study by time since symptom onset and Ct-value category (Table 3), the numbers of participants per stratum were relatively small leading to broad confidence intervals.

COVID-19 infectivity peaks during the period shortly before and after the onset of symptoms when also maximal viral loads in upper respiratory tract material are measured [7, 8]. In this context the test performance for specimen with a qRT-PCR Ct-value beneath 30 was calculated. As this cut off was based on the obtained data, it is to be confirmed by prospective evaluations.

In order to optimize standardization, specimens were transported to the laboratory where the VRD was performed by trained technicians. The final objective is however to perform the VRD at the COVID-19 test centers by MHS personnel. Performance of the test by trained laboratory technicians might overestimate the test accuracy in the definitive clinical setting. Furthermore, in order to perform the VRD at the laboratory, samples were stored and transported on dry ice. Partial destruction of antigen due to freezing could thereby not be excluded and could have resulted in an underestimation of the clinical test sensitivity. Following the study, the routine use of the VRD -with performance of the test at the center on fresh material—was implemented at one MHS test center (Breda). During a follow-up period after this implementation, samples for both qRT-PCR and VRD of 979 individuals were obtained and analyzed. 161 included samples were qRT-PCR positive and 817 qRT-PCR negative. 128 true positive, 2 false positive, 815 true negative, 33 false negative and one uninterpretable VRD result were observed, resulting in an overall clinical sensitivity of 79·5% (95%CI 72·4% -96·8%) and specificity of 99·8% (95%CI 99·1% -100%). Likewise, the clinical sensitivity for samples (n = 132) with a Ct-value beneath 30 93·2% (95%CI 87·5% -96·8%) was comparable with the results found during our study.

The presence of COVID-19 like symptoms is a pre-requisite to be tested at a MHS test-center. As clients make their own appointment through a digital system, we cannot exclude a small number of asymptomatic individuals amongst the included individuals in part one of the study. In part two of the study three asymptomatic subjects were excluded.

Because the high client turn-over at the MHS test-centers (3 minutes per test) could not be compromised, no information on non-participants was gathered. As a consequence, systematic difference with participants could not be excluded.

The current reference standard for diagnosis of an active SARS-CoV-2 infection is qRT-PCR. This highly sensitive and specific test is optimal for the diagnosis of clinically ill patients with a possible indication for treatment and individuals working in or staying at high-risk settings for outbreaks with severe consequences (e.g. long-term care facilities and hospitals). qRT-PCR is however less suited for large scale testing of symptomatic community dwelling individuals for the purpose of disease control. The immense qRT-PCR demand created, greatly stresses microbiologic laboratories and the logistic and administrative challenges intrinsic to qRT-PCR lead to substantial waiting times to get tested and to receive results. Rapid testing and feedback are however essential for control of SARS-CoV-2 community spread [2]. LFA SARS-CoV-2 antigen tests, low-cost, rapid diagnostic tests that can be performed close to the community, could potentially offer an alternative [3, 4].

For subjects tested within 7 days after symptom onset, the negative predictive value was 98% for a test-population with a 20% prevalence. This value increases when the test-population prevalence becomes lower. At the time of writing, a second wave of COVID-19 infections was observed in the Netherlands with a prevalence of 10% to 20% in the test populations. In a questionnaire performed by the Dutch National Institute for Public Health and the Environment (RIVM) amongst 50.000 citizens in June 2020 only 12% of the interviewees that developed symptoms reported to have been tested [9]. When 10% of COVID-19 infected individuals are tested with a 100% sensitive test, 900 in 1000 infected individuals will remain undetected. This strongly supports the use of additional tests with slightly lower sensitivity. We believe the beneficial effect of optimizing test accessibility, as well geographically as in time, on the willingness to get tested will outweigh the limited decrease in test sensitivity by far.

Furthermore, COVID-19 infectivity and viral load in the upper respiratory tract generally peak around the time of symptom onset and decrease gradually during the following days [8, 10]. Infected individuals should be detected in this first timeframe in order to optimize the effect of quarantine measures and contact tracing. For the purpose of COVID-19 control, it is preferential to test early on with suboptimal analytical sensitivity for low viral loads, rather than using a 100% sensitive test only later on in the disease process [2].

In conclusion, the VRD is a promising diagnostic test for testing of symptomatic community-dwelling individuals within seven days after symptom onset for the purpose of disease control. Performance of the test on a large scale is however likely to impose specific logistic challenges. Furthermore, the optimal position of the test within the current testing landscape is to be determined. Further research to practical applicability, appropriate test populations, indications and settings and the potential impact on disease control is needed.

## Supporting information

**S1 Dataset.** S1. Cross tabulation of the BD Veritor System for Rapid Detection of SARS-CoV-2 (VRD) compared for qRT-PC based on visual interpretation of the results (a) and interpretation with the BD Veritor Analyzer (b) excluding Invalid test results (n = 1). S2. Anonymized data of study Part One. S3. Anonymized data of study Part Two.
(XLSX)

## Acknowledgments

We want to thank the employees of the Microvida laboratory for Medical Microbiology at the Amphia Hospital Breda and the MHS West-Brabant for their substantial contribution to this study.

Jan Kluytmans is member of the National Outbreak Management Team of The Netherlands and of a committee which supports the implementation of the Corona-reporting App.

## Author Contributions

**Conceptualization:** Nathalie Van der Moeren, Vivian F. Zwart, Esther B. Lodder, Jan A. J. W. Kluytmans.

**Data curation:** Nathalie Van der Moeren, Vivian F. Zwart, Esther B. Lodder, Wouter Van den Bijllaardt, Harald R. J. M. Van Esch, Joep J. J. M. Stohr, Joost Pot, Ineke Welschen, Petra M. F. Van Mechelen, Suzan D. Pas, Jan A. J. W. Kluytmans.

**Formal analysis:** Nathalie Van der Moeren, Vivian F. Zwart, Joep J. J. M. Stohr, Jan A. J. W. Kluytmans.

**Investigation:** Nathalie Van der Moeren, Vivian F. Zwart, Esther B. Lodder, Wouter Van den Bijllaardt, Harald R. J. M. Van Esch, Joep J. J. M. Stohr, Joost Pot, Suzan D. Pas, Jan A. J. W. Kluytmans.

**Methodology:** Nathalie Van der Moeren, Vivian F. Zwart, Esther B. Lodder, Jan A. J. W. Kluytmans.

**Project administration:** Nathalie Van der Moeren, Vivian F. Zwart.

**Resources:** Nathalie Van der Moeren, Vivian F. Zwart.

**Software:** Nathalie Van der Moeren, Vivian F. Zwart.

**Supervision:** Nathalie Van der Moeren, Vivian F. Zwart, Esther B. Lodder, Wouter Van den Bijllaardt, Joep J. J. M. Stohr, Jan A. J. W. Kluytmans.

**Validation:** Nathalie Van der Moeren, Vivian F. Zwart, Wouter Van den Bijllaardt, Joep J. J. M. Stohr, Suzan D. Pas, Jan A. J. W. Kluytmans.

**Visualization:** Nathalie Van der Moeren, Vivian F. Zwart, Jan A. J. W. Kluytmans.

**Writing – original draft:** Nathalie Van der Moeren, Vivian F. Zwart.

**Writing – review & editing:** Nathalie Van der Moeren, Vivian F. Zwart, Esther B. Lodder, Wouter Van den Bijllaardt, Joep J. J. M. Stohr, Suzan D. Pas, Jan A. J. W. Kluytmans.

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
