## [Decision Letter · Decision Letter 0]

22 Dec 2020

PONE-D-20-36174

Performance evaluation of a SARS-CoV-2 rapid antigen test: test performance in the community in the Netherlands

PLOS ONE

Dear Dr. Nathalie Van der Moeren, 

Thank you for submitting your manuscript to PLOS ONE. After careful consideration, we feel that it has merit but does not fully meet PLOS ONE’s publication criteria as it currently stands. Therefore, we invite you to submit a revised version of the manuscript that addresses the points raised during the review process.

Several major limitations have been highlighted that we would like you to consider in your revision. As per the detailed peer review feedback concerns have been raised including limitations of  the study design presented (combining single cohort design and cases only cohort), better clarity of methods used, better explanation of how the sample size was calculated and how the results and discussions of limitations have been reported. In addition, please recheck the STARD reporting guidelines to check if the relevant items have been reported in your study. Do consider revising your study's title and acknowledging published systematic reviews on the accuracy of SARS-CoV-2 antigen tests in your discussion as the reviewers have recommended. We will appreciate your responses to all the major and minor comments highlighted by the three peer-reviewers.

We look forward to receiving your revised manuscript.

Kind regards,

Eleanor Ochodo, M.D., PhD

Academic Editor

PLOS ONE

2.) Please provide additional details regarding participant consent. In the ethics statement in the Methods and online submission information, please describe how verbal consent was documented and witnessed, and why written consent was not obtained. If your study included minors, state whether you obtained consent from parents or guardians.

3.) To comply with PLOS ONE submission guidelines, in your Methods section, please provide additional information regarding your statistical analyses. In addition, please report your p-values to support your claims. For more information on PLOS ONE's expectations for statistical reporting, please see https://journals.plos.org/plosone/s/submission-guidelines.#loc-statistical-reporting.”

4.) PLOS ONE requires experimental methods to be described in enough detail to allow suitably skilled investigators to fully replicate and evaluate your study. See https://journals.plos.org/plosone/s/submission-guidelines#loc-materials-and-methods for more information. To meet PLOS ONE submission guidelines, in your Methods section, please provide a more detailed description of your RT-qPCR methodology, including the primer sequences used.

5.) We note that you have stated that you will provide repository information for your data at acceptance. Should your manuscript be accepted for publication, we will hold it until you provide the relevant accession numbers or DOIs necessary to access your data. If you wish to make changes to your Data Availability statement, please describe these changes in your cover letter and we will update your Data Availability statement to reflect the information you provide.

6.) Please include a caption for figure 1, 2 and 3.

Reviewers' comments:

Reviewer's Responses to Questions

**Comments to the Author**

1. Is the manuscript technically sound, and do the data support the conclusions?

Reviewer #1: Yes

Reviewer #2: Partly

Reviewer #3: Partly

2. Has the statistical analysis been performed appropriately and rigorously? 

Reviewer #1: Yes

Reviewer #2: No

Reviewer #3: No

3. Have the authors made all data underlying the findings in their manuscript fully available?

Reviewer #1: Yes

Reviewer #2: Yes

Reviewer #3: Yes

4. Is the manuscript presented in an intelligible fashion and written in standard English?

Reviewer #1: Yes

Reviewer #2: No

Reviewer #3: Yes

5. Review Comments to the Author

Reviewer #1: Thank you for the opportunity to review this paper which reports the result of community-based COVID-19 antigen testing in two cohort of symptomatic participants. The study follows good methods for diagnostic test evaluation and on the whole is reported according to the STARD reporting guideline extension for DTA studies. I have identified a couple of areas that the authors may want to consider expanding on, however overall this is a welcome addition to the literature on the potential role of antigen testing for COVID-19.

Pg 7 -Title – could more clearly identify the study as reporting measures diagnostic test accuracy however this is covered in the Abstract so probably sufficient

Pg 8, line 44-49. Abstract – Results – It is not sufficiently clear that the main result presented in the Abstract reports data from a single cohort in part 1 (people presenting for testing at a single centre) combined with a cases only cohort in part 2 (people with a positive PCR who then agreed to have a subsequent second set of swabs for antigen testing and repeat PCR). Combining data in this way creates a sample that is enriched with PCR positive cases, and opens the study to biases associated with diagnostic case control studies with artificially inflated prevalence of disease (see comment below regarding this point). As currently written, the study could be mis-identified as one single gate study that includes all participants meeting criteria for testing.

Pg 10, line 106-7. Methods – Setting – A summary of the criteria for testing in place at the time of the study would be helpful, particular as national criteria for testing may change over time.

Pg 10, line 121-123. Methods - BD Veritor System – Could mention that the manufacturer states that the Analyzer “must be used for interpretation of all test results”.

Pg 11, line 148-149. Methods – Study procedure – Given the evidence for variation in test accuracy according to who obtained the swabs for testing, please describe whether the GGD employees obtaining the swabs were trained health care workers or non health care workers.

Pg 11, line 166-167. Sample size – Should provide some indication of how the target sample sizes were derived.

Pg 12 , line 161-162. Please report whether visual or analyzer based results were used for part 2 of the study

Pg 12, line 182-183. Results – some baseline characteristics of included participants should be reported.

Pg 13, line 202-203. Results and Figure 2. What is of key interest here is the number of participants who had a positive PCR test, the number who were invited to have a home visit, and in particular, the number who consented to the home visit and were tested. It is possible that there are systematic differences between those who agreed to a home visit and those who did not, e.g. they could be older or younger, they may have been more likely to be experiencing continued symptoms or to have more severe symptoms, and this in turn could affect the observed sensitivity of the antigen test. Ideally the authors should report the full participant flow in Figure 2, and tabulate key characteristics according to those who consented or and who did not so that the reader can judge the generalisability of the population.

Pg 14, line 234. Results and Table 3. As per the point made above, I have concerns about combining data for cases in part 1 with part 2 of the study in Table 3, given the different participant flow. I am not sure it adds much to Table 2 anyway, as the pattern in results is the same with only small differences in point estimates and CIs

Pg 14, line 235-239. Results. PPV is only 100% if the results using visual inspection of the assay are used. As mentioned above this is against the manufacturers instructions for use and this needs to be explicit in this sentence.

Pg 14, line 235-239. Results. This paragraph may be difficult for readers to follow without some graphical explanation or tabulation of data. It might be more useful to include this as a replacement for Table 3. I would also suggest using alternative estimates of test sensitivity for calculating NPV. The use of test sensitivity at <7days and Ct <30 appears highly selective to make the data more favourable to the test. In the field, one does not know what an individual’s Ct value will be - some cases of SARS-CoV-2 infection do not achieve high viral loads but that does not necessarily mean that individuals are not infectious. There is also the caveat made above that these results from people agreeing to a second test may not apply more widely. I agree that antigen testing has a role to play for community testing of symptomatic people however suggest the authors take care with over-interpreting the data.

Pg 15-18. Discussion. On the whole I agree with the points made in the discussion and the authors cover the majority of limitations in their data. The results for routine use of Ag testing in the community are interesting and potentially reassuring in regard to the concerns that I have raised about the participants in part 2 of the study.

Pg 17, There are a couple of spelling errors on line 291 - trough instead of through and line 296 our instead of or

Pg 18, line 329-335. Fully agree with the authors conclusions which are in line with the data presented.

Reviewer #2: The aim of this study was to determine the clinical performance of the SARS-CoV-2 rapid antigen test ‘BD Veritor System for Rapid Detection of SARS-CoV-2’. The authors found a high specificity and a reasonable sensitivity. This study is of high importance and clinical relevant in the current pandemic. However, we found the method section unclear and we have concerns about the study designs that were not addressed in the discussion section. They should explain their limitations and the impact of these limitations on the results of the study in the discussion section. Below we give specific comments for the manuscript.

Title

The title includes two times performance. The title could be adjusted to: Clinical performance of a SARS-Cov-2 rapid antigen test in a Dutch community.

Background

- Clear background with a clear description of what is known and what they will add to the literature.

Detailed remarks

- Line 79: qualitative data is not correct as this involves non-numerical data to understand the concepts and opinions. Maybe the authors mean quality data? The term qualitative should be removed. (see also line 85 and 118)

- Line 79: It is unclear what is meant by performance. The authors determine the diagnostic accuracy (sensitivity, specificity, negative and positive predictive value). It is more clear when the authors uses the term diagnostic accuracy (or clinical performance) and specify this in the introduction.

- Line 84 to 87: sentence is not clear and should be rewritten

- Line 87: GGD is a Dutch abbreviation and I am not sure if this is known in other countries. Therefore, use the abbreviation of MHS (Municipal Health Service) or explain the Dutch abbreviation GGD (Gemeentelijke Gezondheids Dienst).

Methods

- Major problem is that they did not use one large prospective cohort study in which they could determine the sensitivity and specificity in the same cohort. The second cohort in the study was based on the selection of confirmed cases, which could overestimate the sensitivity of the test. In addition, the sample size was not specified. How did you determine to include 300 negatives for part one and 100 positives for part two of the study? The following reference could be used for determining the sample size: Buderer NM. Statistical methodology: I. Incorporating the prevalence of disease into the sample size calculation for sensitivity and specificity. Acad Emerg Med 1996;3:895e900.

- Explain your choices in the methods in more detail. On what criteria were the various Ct values groups based, why did you used 7 days as a threshold while the manufacturer uses 5 days after disease onset, why were only two of the three test centres included for this study (or was only Breda included)?

- The two or three parts of the study are not very clear throughout the manuscript and the text on these parts are not consistent throughout the method section. For example, what will be determined in each part and at which center are the patients recruited.

- Why were the VRD performed by trained laboratory technicians and not by the GGD personal. Trained laboratory technicians could make fewer mistakes compared to the GGD personal, which will overestimate the sensitivity and specificity in clinical practice.

Detailed remarks

- Line 98: explain abbreviation Ct-value (cycle threshold) when this is used for the first time in your manuscript.

- Line 106: second time that GGD/MHS abbreviation is explained. This is unnecessary as this is already specified in the introduction.

- Line 106: What were the COVID-19 like symptoms?

- Line 110: BCO (Bron contact Onderzoek) is also a Dutch abbreviation. Change this to the English abbreviation.

- Line 134 and 139: Was verbal informed consent enough? Was a written informed consent not necessary?

- Line 148: reference of Dutch national COVID-19 test protocol is missing.

- Line 170: part three of the study is not explained before and it is unclear what is meant by this.

- Line 174: why was the NPV and PPV not calculated with the prevalence of study one (2x2 table), which represent the prevalence in the patients suspected of COVID-19 presenting at the GGD/MHS? This could be added to table 1.

- Line 176: explain abbreviation VA.

Results

- A table with baseline characteristics of the patient population is missing. This could provide insight in the tested population (e.g. what were the COVID-19 like symptoms)

- The subgroups in table 2 had very small numbers, resulting in large 95% confidence intervals. This should be mentioned in the discussion section.

- The number of patients included in part two is not clear: Figure 2 mentioned 129, table 2 mentioned 123 and the abstract also mentioned 123.

- Why were part one and two combined? This is not explained in the method section.

Detailed remarks

- Figure 1: how many patients were asked to participate in the study and did not give informed consent?

- Line 203: mention that the three asymptomatic patients were excluded from to analysis of part two and explain in the methods why you excluded these patients.

- Table 2: the abbreviations used were not explained underneath the table.

- Line 239: 95% confidence intervals of PPV and NPV are missing. In addition, it is not clear how NPV was calculated and this should be explained in the method section.

Discussion

- They mention that the literature on clinical performance is scares, however there is a Cochrane review on point of care tests for SARS-CoV-2 infection that included 22 publications and this should be discussed in the discussion section: Dinnes, J., Deeks, J.J., Adriano, A., Berhane, S., Davenport, C., Dittrich, S., Emperador, D., Takwoingi, Y., Cunningham, J., Beese, S. and Dretzke, J., 2020. Rapid, point‐of‐care antigen and molecular‐based tests for diagnosis of SARS‐CoV‐2 infection. Cochrane Database of Systematic Reviews, (8).

- They mention that strengths of the study are the prospective design and large sample size. However, only part one had a prospective design, while part two had more of a case-control design that could overestimate the sensitivity. In addition, they did not include one large prospective cohort. When a sample size calculation would be performed according to Buderer 1996, slightly more than 1000 patients would be needed. Therefore, they cannot argue that they include a large number of patients. In addition, the number of patients in the subgroup analysis were very small resulting in low precision.

- RT-PCR is not the gold standard, but more a reference standard. There are also guidelines and clinical presentations used as a reference standard. This issue could be discussed in the discussion section.

Detailed remarks

- Line 246: It is not common to refer to a figure in the discussion section.

- Line 282 to 289: it is not common to include new results of a follow-up study in the discussion section. Why is this not included in the methods and results section?

- Line 294 to 308: This is repetition of information that was also stated in the introduction and could be deleted.

- Line 315 to 322: The use of a study performed in the United States is not clear, as the situation is quite different in the United States compared to the Netherlands.

Reviewer #3: The authors report a diagnostic performance study comparing SARS-CoV-2 rapid antigen testing to batched laboratory RT-PCR testing using 2 commercial PCR assays. The study reports on a high prevalence setting in a symptomatic population presenting to an outpatient testing facility in the Netherlands in the early fall of 2020. Two cohorts are used, one prospectively collected, blinded for the outcome and one cohort assessing test characteristics in prior test positives only. They also aim to report on agreement between the application of an automated reader compared to manually reading of the test result.

The manuscript is in general replicable and reasonably well described. It however combines a technically very sound study design with a more flawed second arm (positives only) and the second agreement rates are not quantified. It is unclear on which basis the sample size is calculated and this sample size remains rather small.

The main comment is the duality of the study question and objective: it is unclear if the authors try to answer a more population health testing in a clinical sick population (which this study cannot) or can agree that their data are those from a clinical diagnostic setting and the conclusions should remain focused to that extend (as clearly stated in their objective section – but circled around both in the abstract, introduction and discussion section). A further elaboration on counterfactual further implications is possible, but should be left for the discussion section only to be appropriate.

However, diagnostic performance studies on test positives and negatives remain rare, more high quality quantitative data are necessary, so I thank the authors for the effort of writing up their findings.

Major comments:

-The article turns back and forward between the antigen test being a test for clinical practice vs for community testing. Community testing, at this point, has however rather the connotation of screening for infection and infectiousness and being used as a public health intervention tool (non-pharmaceutical intervention). This is not what the study is investigating. This study looks at the performance in a cohort of ill and symptomatic individuals to make or refute the diagnosis of SARS-CoV-2 infection and COVID-19 disease.

There is lack of evidence of the use of antigen testing both in a (primary care) clinical setting and in a public health focused context. This study is performed in a population with a prevalence of 29% and all patients are or are supposed to be symptomatic and thus not a representative nor generalizable population for the population health test setting.

The objective is clearly stated as to assess clinical spec and sens – for disease diagnosis amongs symptomatic individuals. How the data can be interpreted in the broader picture of community testing in the PH sense would best be kept for in the discussion (and not in the background – introduction – of neither the manuscript nor the abstract – given it gives a false expectation of the setting and results)– given this study is limited in its evidence to that extend. In addition, the patient group that was assessed for, as is mentioned, the sens, is clinically a group further away from how a rapid tests would be applied in the public health setting – given those individuals are already at the further right side of their infection time-curve. This study thus gives mainly data regarding this right tail of infected and symptomatic individuals.

-What is meant with clinical sensitivity is not completely clear.

-A reference to the symptoms that were accepted/suggested to being tested in the region at the time of testing is informative and can be added in the references.

-Data on the patient population characteristics are missing: any age distribution?

-A cut off for timing was 7 days after onset of symptoms: there is no explanation why 7 days was chosen – given this categories a continuous variable into a binary variable and leads to loss of information, at minimum a reason for this choice should be made (this can be based on evidence on infectiousness or Ct evolution and discussed with your Fig 3).

-The specimen sample type is mentioned in the manuscript – but preferably also has a place in the abstract, mainly given the importance of specimen type in the accuracy of diagnostic tests.

-The STARD checklist should be completed. Major comments on the elements that are part of STARD are (additional others are integrated in the detailed comments):

*Incomplete reporting of elements in the title

*Introduction: the intended use and clinical role are, as mentioned prior, mixed and it is best to improve clarity.

*Reference standard misses detail, as well participant description

Detailed comments:

Title:

It is key that this was investigated in symptomatic individuals – and this information is best included in the title. The term “community testing” is a bit misleading – given it is in a primary care setting that was serving clinical diagnosis. Mainly because already in the introduction community testing is described as a general testing strategy.

Abstract:

-“Application for large- scale community testing for disease control purposes “ This is also not what this article is about – rather a place in the discussion.

-“ to qRT-PCR “: best to add if in house or a commercial assay.

-What is low Ct value: best to give cut off

-what is magical about 7 days: is this 7 days because days since onset of symptoms was investigated as a continuous variable? (to add in text)

Background & Methods:

-line 65: to establish COVID-19 infection: best to add “acute” and is not COVID-19 infection, but SARS-CoV-2 infection, given COVID-19 is the disease.

-line 69: the PCR technology is not limited to specialized laboratories – the batched testing is, however point-of-care PCR instruments and assays are available – when receiving a CLIA waiver they can be used close to the patient. The sample can be precurred everywhere – even by self-sampling. It is the whole cascade with mostly high-throughput batched PCR and non-self sampling that do not allow for broader community based testing and that is not focused on clinical diagnosis but on community screening in the context of a public health intervention that is referred to but not the real setting of your work.

-line 72: the word “pressurizes”: I understand the meaning, but it is not a term used as such: better to replace by “stresses or burdens”.

-line 74: rapid testing: rapid testing and reporting (if you indeed want the whole circle to be happening fast).

-line 79: what is meant with qualitative? (might be literal translation) What is probably meant and needed: high quality quantitative data

-line 80: missing “the” in front of clinical setting

-line 86: “respiratory specimen” is rather broad: at least upper resp tract specimen and better to name which specimen type is used (saliva vs nose vs throat vs NPS…): This is relevant information that needs to be mentioned here in this last paragraph where the objective and PICOT of the study is stated. (it does come back in your manuscript later, which is appreciated).

-line 98: what is means with clinical sens for the Ct values? How is clinical sens here defined?

-line 108: missing a prior to specimen

-line 112: an additional word about: testing strategy that was in place at that moment; is this rural – city? – population catchment area?

-line 120: the manufacturer does not report a CI?

-line 124: check English expression: visible to the naked eye

-line 128: the reference standard is insufficiently described. The described platforms are not assays – a platform cannot have a target. On a platform one can use an assay – it is thus necessary to name the assay. Additional: reference to where the assay is described – approved by authorities? – some reference to its performance – mainly if validated for the specimen type and performance for the in the study used specimen collection process.

-line 151: one time “in” too many

-line 166-167: when there is mentioning of the sample size: why was this sample size chosen if such a specific number. To elaborate.

-line 175: “for a range of prevalence” : this is the methods section, so one can as well be precise and quote the range. As well: it is stated that agreement is assessed: using which methods? The analysis section misses the methods used to assess agreement and the calculations. More info on the methods is needed here. How are CI calculated?

Results

-line 184: re-write - more literal translation

-line 189: thank you to report on uninterpretable data – important to mention and that this proportion was very small.

-line 209: Best to refer to: PCR pos - the pos rate by VRD was x among PCR positives.

-line 224: it is unwarranted and incorrect to merge the 2 cohorts and give the PPV and NPV – which merges 2 groups with a very different pre-test probability – which, and thank you to report is unrealistically high: 60%. (latter comment on line 234 and on).

-line 225: plural of specimen is specimens

-line 239: here it is clear that you are assessing its performance in a high prevalence setting – thus clinical setting – because the prevalence reported on is still too high for general screening, where it will be lower than 10%. The NPV of course will be even more improved, given very low prevalence.

Discussion:

-line 264: the prospective arm here is overstated. It remains unclear if this is your target setting or not.

-line 273: “As this cut off was based…”: unclear what the message is of this sentence.

-line 282: A key element to mention is if the selection criteria to be tested were the same at the Breda sample in this later period – where we know that testing criteria have shifted over time. This discussion and clarity about test criteria might have a better place earlier in the discussion.

-line 319: What is written supports the opportunity and the counterfactual of: with more tests, even with lower sens, this strategy will capture an absolute total of infected individuals that is larger compared to those now. It is appreciated that this is now discussed purely in the scheme of the symptomatic non-tested individuals – where this manuscript has the data to support its use in that specific group (compared to not having data to support PH community screening). This sentence might need some re-formulation however.

-line 326-328: is there data to back this up? Reference? At least some more elaborated discussion – reference – opinion – study – even modelling…

-line 330: this conclusion is not warranted that this test has – had an impact on disease control – the study did not look at this outcome in any way – but assumingly this is why it is written that it is promising – but the conclusion of your study should rather be based on what your data support. Rather as well a proposal on how further research can be meaningful and feasibly be performed and implemented can be of added value.

-line 331: Performance: might it be that rather is meant “Implementation of the test”…

Tables and figures:

-Table 1: this is not a table that shows a real comparison – given it purely lists the effect estimates – it is not a real agreement evaluation neither. Cross tabulation is necessary.

-Figure 1: the legend and info below a flow diagram and all tables and figures should be complete and self-explanatory. Part one better to be replaced by prospective cohort… (readers might be only looking at the graphs). As well: potentially eligible: they were or they were not – their status of being eligible was not potential: it was real. Or what is the potential of being eligible?

6. PLOS authors have the option to publish the peer review history of their article (what does this mean?). If published, this will include your full peer review and any attached files.

Reviewer #1: **Yes: **Dr Jac Dinnes

Reviewer #2: **Yes: **Gea A. Holtman

Reviewer #3: **Yes: **Joanna Merckx

---

## [Author Response · Author response to Decision Letter 0]

12 Feb 2021

Dear Editor,

We would like to thank you for the thorough evaluation of our manuscript 'Performance evaluation of a SARS-CoV-2 rapid antigen test : test performance in the community in the Netherlands' (PONE-D-20-36174) and the opportunity to resubmit a revised copy. We are very grateful for the valuable comments and feedback from the editor and reviewers, which we believe have resulted in a greatly improved revised manuscript. 

Please find the responses to the points raised by the academic editor and the reviewers - following the original comment in italics - beneath. Two versions of the revised manuscript, one with and one without track changes, were uploaded alongside with this document. 

Thank you for your consideration of our revised manuscript.

Sincerely yours,

Nathalie Van der Moeren

on behalf of the co-authors 

Points raised by the Academic Editor

1. Please ensure that your manuscript meets PLOS ONE's style requirements,

including those for file naming. 

The manuscript was adapted in accordance with the PLOSONE style requirements.

2. Please provide additional details regarding participant consent. In the ethics statement in the Methods and online submission information, please describe how verbal consent was documented and witnessed, and why written consent was not obtained. If your study included minors, state whether you obtained consent from parents or guardians.

In the first part of the study, individuals were informed about the study trough local media, by MHS communication channels (full participant information letter on the website, …) and by information signs at the participating test centres. Verbal informed consent was obtained separately by two independent MHS employees. No written informed consent was obtained as this would have compromised the strictly needed high flow of individuals being tested at the test centres. (3 minutes per client). In the second part of the study, potential participants were informed about the study and asked for verbal informed consent a first time by telephone. Verbal informed consent was obtained by a different MHS employee a second time during a home visit before obtainment of the study samples. No written informed consents were obtained as handling of documents obtained from confirmed infectious participants was considered a potential safety hazard. 

This information was added to the manuscript (line 157-167)

No minors were included as stated in the paragraph ‘Patient recruitment’. (line 143) 

3. To comply with PLOS ONE submission guidelines, in your Methods section, please provide additional information regarding your statistical analyses. In addition, please report your p-values to support your claims. For more information on PLOS ONE's expectations for statistical reporting, please see https://journals.plos.org/plosone/s/submission-guidelines.#loc-statistical-reporting.”

Thank you for this valuable remark, we elaborated on the statistical analysis used (lines 200-211) and added p-values to support our claims (lines 257-259, 262)

4. PLOS ONE requires experimental methods to be described in enough detail to allow suitably skilled investigators to fully replicate and evaluate your study. See https://journals.plos.org/plosone/s/submission-guidelines#loc-materials-and-methods for more information. To meet PLOS ONE submission guidelines, in your Methods section, please provide a more detailed description of your RT-qPCR methodology, including the primer sequences used. 

Thank you for this valid remark, we elaborated on the RT-qPCR methods used. (line 135-140) As commercial kits were used and primer sequences are propriatory information of the manufacturer, we are not able to provide more information of the primer sequences used.

We would like to change our Data Availability statement. We would like to add the anonymised data to the manuscript as two supplementary tables, which were uploaded alongside with this ‘Responses to the Authors’ letter.

6. Please include a caption for figure 1, 2 and 3.

The captions for figure 1,2 and 3 were added to the manuscript at the bottom of the appropriate paragraphs. 

Points raised by Reviewer 1

1. Pg 7 -Title – could more clearly identify the study as reporting measures diagnostic test accuracy however this is covered in the Abstract so probably sufficient.

The title was adjusted (Line 1-2) 

2. Pg 8, line 44-49. Abstract – Results – It is not sufficiently clear that the main result presented in the Abstract reports data from a single cohort in part 1 (people presenting for testing at a single centre) combined with a cases only cohort in part 2 (people with a positive PCR who then agreed to have a subsequent second set of swabs for antigen testing and repeat PCR). Combining data in this way creates a sample that is enriched with PCR positive cases, and opens the study to biases associated with diagnostic case control studies with artificially inflated prevalence of disease (see comment below regarding this point). As currently written, the study could be mis-identified as one single gate study that includes all participants meeting criteria for testing.

We are grateful for this very valuable remark. We clarified the nature and existence of the two cohorts in the abstract and removed the part where both cohorts were combined from the abstract, methods and results section. 

3. Pg 10, line 106-7. Methods – Setting – A summary of the criteria for testing in place at the time of the study would be helpful, particular as national criteria for testing may change over time.

Individuals can – provided they state to have COVID-19 like symptoms (rhinitis, cough, elevated temperature (not further specified), shortness of breath or sudden loss of sense of taste or smell) - make an appointment at a regional MHS test centre. These criteria remained unchanged during the whole study period. This information was added to the manuscript (line 108-112) 

4. Pg 10, line 121-123. Methods - BD Veritor System – Could mention that the manufacturer states that the Analyzer “must be used for interpretation of all test results”.

The sentence stating the manufacturer instruction to use the reader was adapted to ‘The manual prescribes interpretation of the results after 15 minutes with a reading device provided by the manufacturer (VA).’ (line 132-133) 

5. Pg 11, line 148-149. Methods – Study procedure – Given the evidence for variation in test accuracy according to who obtained the swabs for testing, please describe whether the GGD employees obtaining the swabs were trained health care workers or non health care workers.

The GGD employees were specifically trained to obtain nasopharyngeal samples, but were as a rule no trained healthcare workers. This information was added to the manuscript (line 113-114). 

6. Pg 11, line 166-167. Sample size – Should provide some indication of how the target sample sizes were derived.

We would like to thank the reviewer for this valuable remark, the methods used to determine the sample size were added to the manuscript. (line 195-197)

7. Pg 12 , line 161-162. Please report whether visual or analyzer based results were used for part 2 of the study. 

In part two of the study only results of the visual interpretation were used, this was added to the manuscript. (line 190) 

8. Pg 12, line 182-183. Results – some baseline characteristics of included participants should be reported.

For part one of the study the included participants were men and women aged 18 years and above, we do unfurtunately not have any more demographic data available. For part two of the study the participants ages varied from 18 to 84 years (M = 44, SD=16). The available demographic data was added to the manuscript.(line 215 and 251-252)

9. Pg 13, line 202-203. Results and Figure 2. What is of key interest here is the number of participants who had a positive PCR test, the number who were invited to have a home visit, and in particular, the number who consented to the home visit and were tested. It is possible that there are systematic differences between those who agreed to a home visit and those who did not, e.g. they could be older or younger, they may have been more likely to be experiencing continued symptoms or to have more severe symptoms, and this in turn could affect the observed sensitivity of the antigen test. Ideally the authors should report the full participant flow in Figure 2, and tabulate key characteristics according to those who consented or and who did not so that the reader can judge the generalisability of the population.

We are thankful for this valid remark, unfortunately however this information was not gathered as the high turnover of clients at the MHS test centres (1 test every 3 minutes) could not be compromised. The lack of data on non-participants was added as a limitation to the discussion section of the article. (line 333-335)

10. Pg 14, line 234. Results and Table 3. As per the point made above, I have concerns about combining data for cases in part 1 with part 2 of the study in Table 3, given the different participant flow. I am not sure it adds much to Table 2 anyway, as the pattern in results is the same with only small differences in point estimates and Cis

We are grateful for this justified remark, the section on the combined data (including table 3) was deleted from the abstract, methods, results and discussion section. 

11. Pg 14, line 235-239. Results. PPV is only 100% if the results using visual inspection of the assay are used. As mentioned above this is against the manufacturers instructions for use and this needs to be explicit in this sentence.

This information was emphasized in the manuscript. (line 275) 

12. Pg 14, line 235-239. Results. This paragraph may be difficult for readers to follow without some graphical explanation or tabulation of data. It might be more useful to include this as a replacement for Table 3. I would also suggest using alternative estimates of test sensitivity for calculating NPV. The use of test sensitivity at <7days and Ct <30 appears highly selective to make the data more favourable to the test. In the field, one does not know what an individual’s Ct value will be - some cases of SARS-CoV-2 infection do not achieve high viral loads but that does not necessarily mean that individuals are not infectious. There is also the caveat made above that these results from people agreeing to a second test may not apply more widely. I agree that antigen testing has a role to play for community testing of symptomatic people however suggest the authors take care with over-interpreting the data.

We would like to thank the reviewer for this helpful comment. We added a table with the PPV and NPV (table 2) and used the overall test sensitivity estimate from Part one of the study instead of the sensitivity based on the combined data from part three. (line 237-244)

13. Pg 15-18. Discussion. On the whole I agree with the points made in the discussion and the authors cover the majority of limitations in their data. The results for routine use of Ag testing in the community are interesting and potentially reassuring in regard to the concerns that I have raised about the participants in part 2 of the study.

We would like to thank the reviewer for this remark. 

14. Pg 17, There are a couple of spelling errors on line 291 - trough instead of through and line 296 our instead of or

Thank you, corrections were made. 

15. Pg 18, line 329-335. Fully agree with the authors conclusions which are in line with the data presented.

We would like to thank the reviewer for this positive feedback. 

Points raised by Reviewer 2

The aim of this study was to determine the clinical performance of the SARS-CoV-2 rapid antigen test ‘BD Veritor System for Rapid Detection of SARS-CoV-2’. The authors found a high specificity and a reasonable sensitivity. This study is of high importance and clinical relevant in the current pandemic. However, we found the method section unclear and we have concerns about the study designs that were not addressed in the discussion section. They should explain their limitations and the impact of these limitations on the results of the study in the discussion section. Below we give specific comments for the manuscript.

1. The title includes two times performance. The title could be adjusted to: Clinical performance of a SARS-Cov-2 rapid antigen test in a Dutch community.

The title was adjusted according to this remark and the points raised by reviewer one and three. (line 1-2) 

2. Clear background with a clear description of what is known and what they will add to the literature.

We would like to thank the reviewer for this positive feedback.

3. Line 79: qualitative data is not correct as this involves non-numerical data to understand the concepts and opinions. Maybe the authors mean quality data? The term qualitative should be removed. (see also line 85 and 118)

Thank you for this helpful remark, the sentences were adapted accordingly. (line 35,79,283) 

4. Line 79: It is unclear what is meant by performance. The authors determine the diagnostic accuracy (sensitivity, specificity, negative and positive predictive value). It is more clear when the authors uses the term diagnostic accuracy (or clinical performance) and specify this in the introduction.

The term performance was replaced by test accuracy. (line 1, 37, 85, ..)

5. Line 84 to 87: sentence is not clear and should be rewritten

The sentence was adapted. (line 85-88)

6. Line 87: GGD is a Dutch abbreviation and I am not sure if this is known in other countries. Therefore, use the abbreviation of MHS (Municipal Health Service) or explain the Dutch abbreviation GGD (Gemeentelijke Gezondheids Dienst).

The abbreviation GGD was replaced by MHS throughout the manuscript.

7. Major problem is that they did not use one large prospective cohort study in which they could determine the sensitivity and specificity in the same cohort. The second cohort in the study was based on the selection of confirmed cases, which could overestimate the sensitivity of the test. 

We are grateful for this justified remark. We deleted the part in which the results of part one and two of the study were combined and clarified the characteristics of the two cohorts with specific objectives in the abstract and the methods section. 

8. In addition, the sample size was not specified. How did you determine to include 300 negatives for part one and 100 positives for part two of the study? The following reference could be used for determining the sample size: Buderer NM. Statistical methodology: I. Incorporating the prevalence of disease into the sample size calculation for sensitivity and specificity. Acad Emerg Med 1996;3:895e900.

We would like to thank the reviewer for this valuable remark, the methods used to determine the sample size were different from the table stated in Budere et al and were added to the manuscript. (line 195-197)

9. Explain your choices in the methods in more detail. On what criteria were the various Ct values groups based, why did you used 7 days as a threshold while the manufacturer uses 5 days after disease onset, why were only two of the three test centres included for this study (or was only Breda included)?

The cut-off of 7 days was based on the results of Bullard et al. (Bullard, J. et al. Predicting infectious SARS-CoV-2 from diagnostic samples. Clin. Infect. Dis. 71, 2663–2666 (2020)) that showed no viral growth after incubation on Vero cells in positive samples obtained more than 8 days after symptom onset. This information was added to the manuscript. (line 206)

Only one test centre was included in part one of the study because of logistic reasons (logistic set up in one place, distance from the laboratory to the test centre), in the second part of the study all qRT-PCR positive samples from test centres in the region were included provided that the qRT-PCR was performed in a Microvida laboratory. This was clarified in the manuscript. (line 120-124)

10. The two or three parts of the study are not very clear throughout the manuscript and the text on these parts are not consistent throughout the method section. For example, what will be determined in each part and at which centre are the patients recruited.

Thank you for this valuable remark. Part three of the study (combined data) was deleted from the manuscript. The cohorts and objectives of part one and two were clarified in het abstract and method section. 

We clarified in which centres patients were recruited (line 120-124)

11. Why were the VRD performed by trained laboratory technicians and not by the GGD personal. Trained laboratory technicians could make fewer mistakes compared to the GGD personal, which will overestimate the sensitivity and specificity in clinical practice.

In consultation with the MHS there was chosen to use trained laboratory technicians at the start of the practical implementation of the CRD at the MHS test centres. This is why trained laboratory technicians were chosen in the study. As in time the goal would be to have the test performed by MHS personnel, this is however a valuable point, it was added as a limitation to the discussion section. (line 314-316) 

12. Line 98: explain abbreviation Ct-value (cycle threshold) when this is used for the first time in your manuscript. 

The explanation of the abbreviation was added to the manuscript. (line 43)

13. Line 106: second time that GGD/MHS abbreviation is explained. This is unnecessary as this is already specified in the introduction.

The second explanation of the abbreviation was removed from the manuscript. 

14. Line 106: What were the COVID-19 like symptoms?

In the MHS guidelines these are described as rhinitis, cough, elevated temperature (not specified), shortness of breath or sudden loss of sense of taste or smell. This information was added to the manuscript. (line 109)

15. Line 110: BCO (Bron contact Onderzoek) is also a Dutch abbreviation. Change this to the English abbreviation. 

The abbreviation was deleted from the manuscript. 

16. Line 134 and 139: Was verbal informed consent enough? Was a written informed consent not necessary?

For part one of the study verbal informed consent was obtained separately by two independent MHS employees. Written informed consent could not be obtained as this would have compromised the strictly needed high flow of individuals being tested at the test centres (3 minutes per client). In the second part of the study, potential participants were asked for verbal informed consent a first time by telephone and a second time before obtainment of the study sample. No written informed consents were obtained as handling of documents obtained from confirmed infectious participants was considered a potential safety hazard. 

The protocol as such was granted an exemption of the Dutch medical scientific research act (WMO).

This information was also added to the manuscript (line 158-168) 

17. Line 148: reference of Dutch national COVID-19 test protocol is missing.

The reference number of the Netherlands Trial Register (NL9018) was added to the manuscript. (line 156-15)

18. Line 170: part three of the study is not explained before and it is unclear what is meant by this. 

Thank you for this valuable remark, references to part three of the study (combined data part one and two) were removed from the manuscript (see also above) 

19. Line 174: why was the NPV and PPV not calculated with the prevalence of study one (2x2 table), which represent the prevalence in the patients suspected of COVID-19 presenting at the GGD/MHS? This could be added to table 1.

Thank you for this valid and helpful remark, NPV and PPV for the prevalence found in cohort one (4.8%) was added. NPV and PPV for a population prevalence of 10 and 20% was calculated based on the sensitivity and specificity estimates found in part one of the study (line 237-241) 

20. Line 176: explain abbreviation VA.

Thank you for this remark, the abbreviation was however already explained in lines 100-101 as ‘reading device provided by the manufacturer’.

21. A table with baseline characteristics of the patient population is missing. This could provide insight in the tested population (e.g. what were the COVID-19 like symptoms)

For part one of the study the included participants were men and women aged 18 years and above, we do unfortunately not possess any more demographic data. For part two of the study the participants ages varied from 18 to 84 years (M = 44, SD=16). The available demographic data was added to the manuscript.(line 215 and 251-252)

22. The subgroups in table 2 had very small numbers, resulting in large 95% confidence intervals. This should be mentioned in the discussion section.

Thank you for this valid remark, the point was added to the limitations in the discussion section. (line 305-307) 

23. The number of patients included in part two is not clear: Figure 2 mentioned 129, table 2 mentioned 123 and the abstract also mentioned 123.

129 symptomatic participants were included of which 123 still had a positive qRT-PCR at the moment of the second sampling. This was emphasized in the table caption and the result section. (line 247-250) 

24. Why were part one and two combined? This is not explained in the method section 

We are grateful for this valid and valuable remark, part three of the study was removed from the manuscript (abstract, methods, results and discussion) 

25. Figure 1: how many patients were asked to participate in the study and did not give informed consent? 

Unfortunately, this information was not gathered as the high turnover of clients at the MHS test centres (1 test every 3 minutes) could not be compromised. We realise this is a limitation of our study, the lack of data on non-participants was added to the limitations in the discussion section. (line 333-335)

26. Line 203: mention that the three asymptomatic patients were excluded from to analysis of part two and explain in the methods why you excluded these patients.

Being or haven been symptomatic was an inclusion criterium for part two of the study, this was clarified in the method section (line 150)

27. Table 2: the abbreviations used were not explained underneath the table.

The explanations of the abbreviations were added underneath the table. 

28. Line 239: 95% confidence intervals of PPV and NPV are missing. In addition, it is not clear how NPV was calculated and this should be explained in the method section.

The PPV and NPV were calculated using Medcalc, this information and 95% CI for NPV were added to the manuscript. (line 204) 

29. They mention that the literature on clinical performance is scares, however there is a Cochrane review on point of care tests for SARS-CoV-2 infection that included 22 publications and this should be discussed in the discussion section: Dinnes, J., Deeks, J.J., Adriano, A., Berhane, S., Davenport, C., Dittrich, S., Emperador, D., Takwoingi, Y., Cunningham, J., Beese, S. and Dretzke, J., 2020. Rapid, point‐of‐care antigen and molecular‐based tests for diagnosis of SARS‐CoV‐2 infection. Cochrane Database of Systematic Reviews, (8).

We discussed the results of Dinnes et al. in the discussion section of the original manuscript (reference 5, lines 406-413). The review in question only included 5 studies on rapid antigen tests, the remaining studies were on rapid molecular tests which we found were out of the scope of the discussion of our manuscript. Furthermore, the included studies were often performed on remnant specimen stored in virus transport medium and often contained little information on days since disease onset and the clinical setting they were obtained in. We changed the sentences stating ‘literature is scarce’ to ‘quality literature is limited’. (lines 36, 284,..)

30. They mention that strengths of the study are the prospective design and large sample size. However, only part one had a prospective design, while part two had more of a case-control design that could overestimate the sensitivity. In addition, they did not include one large prospective cohort. When a sample size calculation would be performed according to Buderer 1996, slightly more than 1000 patients would be needed. Therefore, they cannot argue that they include a large number of patients. In addition, the number of patients in the subgroup analysis were very small resulting in low precision.

We are grateful for this valid remark. The differences in design between part one and two of the study were emphasized throughout the manuscript (see above). The discussion paragraph on strengths of the study was limited to ‘The prospective design of part one of the study and and the obtainment of samples in the target setting of potential use are great assets of this study.’ (line 296)

The valuable remark on the small numbers of the participants in the strata was also addressed in the discussion section (line 305-307)

31. RT-PCR is not the gold standard, but more a reference standard. There are also guidelines and clinical presentations used as a reference standard. This issue could be discussed in the discussion section.

Thank you for this valuable remark, adjustments were made in the manuscript. (line 130, 333) 

32. Line 246: It is not common to refer to a figure in the discussion section.

The reference to the figure was removed from the discussion. 

33. Line 282 to 289: it is not common to include new results of a follow-up study in the discussion section. Why is this not included in the methods and results section?

After implementation, there was a short period in which qRT-PCR were performed alongside the VRD. This was not in a study setting, but a way to monitor the performance in the first period after implementation. 

34. Line 294 to 308: This is repetition of information that was also stated in the introduction and could be deleted.

Thank you for this remark, we substantially shortened the paragraph. 

35. Line 315 to 322: The use of a study performed in the United States is not clear, as the situation is quite different in the United States compared to the Netherlands.

The reference was deleted from the manuscript. 

Points raised by Reviewer 3

The authors report a diagnostic performance study comparing SARS-CoV-2 rapid antigen testing to batched laboratory RT-PCR testing using 2 commercial PCR assays. The study reports on a high prevalence setting in a symptomatic population presenting to an outpatient testing facility in the Netherlands in the early fall of 2020. Two cohorts are used, one prospectively collected, blinded for the outcome and one cohort assessing test characteristics in prior test positives only. They also aim to report on agreement between the application of an automated reader compared to manually reading of the test result.

The manuscript is in general replicable and reasonably well described. It however combines a technically very sound study design with a more flawed second arm (positives only) and the second agreement rates are not quantified. It is unclear on which basis the sample size is calculated and this sample size remains rather small.

The main comment is the duality of the study question and objective: it is unclear if the authors try to answer a more population health testing in a clinical sick population (which this study cannot) or can agree that their data are those from a clinical diagnostic setting and the conclusions should remain focused to that extend (as clearly stated in their objective section – but circled around both in the abstract, introduction and discussion section). A further elaboration on counterfactual further implications is possible, but should be left for the discussion section only to be appropriate.

However, diagnostic performance studies on test positives and negatives remain rare, more high quality quantitative data are necessary, so I thank the authors for the effort of writing up their findings.

We would like to thank the reviewer for this general comment. We will elaborate on the concerns on ‘the duality of the study question’ in the response to comment 1. 

1. The article turns back and forward between the antigen test being a test for clinical practice vs for community testing. Community testing, at this point, has however rather the connotation of screening for infection and infectiousness and being used as a public health intervention tool (non-pharmaceutical intervention). This is not what the study is investigating. This study looks at the performance in a cohort of ill and symptomatic individuals to make or refute the diagnosis of SARS-CoV-2 infection and COVID-19 disease.

The target group of the MHS test centres are individuals who are symptomatic, but do not require hospitalisation or attendance by a physician. The goal is in other words to detect infectious individuals, promptly quarantine them and start contact tracing. This is why the tests are performed by the Dutch MHS. Testing of clinically ill patients who need medical attention is the responsibility of family doctors and hospitals. The MHS test centres are in other words a public health intervention tool and not a centre for the clinical diagnosis of patients in order to direct treatment. 

2. There is lack of evidence of the use of antigen testing both in a (primary care) clinical setting and in a public health focused context. This study is performed in a population with a prevalence of 29% and all patients are or are supposed to be symptomatic and thus not a representative nor generalizable population for the population health test setting.

The objective is clearly stated as to assess clinical spec and sens – for disease diagnosis amongs symptomatic individuals. How the data can be interpreted in the broader picture of community testing in the PH sense would best be kept for in the discussion (and not in the background – introduction – of neither the manuscript nor the abstract – given it gives a false expectation of the setting and results)

given this study is limited in its evidence to that extend. In addition, the patient group that was assessed for, as is mentioned, the sens, is clinically a group further away from how a rapid tests would be applied in the public health setting – given those individuals are already at the further right side of their infection time-curve. This study thus gives mainly data regarding this right tail of infected and symptomatic individuals.

We would like to thank the reviewer fort this remark. The study part with combined data was deleted from the manuscript, resulting in one prospective cohort in part one of the study with a prevalence of 4.8% and a second cohort in part two with known qRT-PCR positive participants only. We think it is a valuable comment that the individuals from the latter cohort are more on the right tale of the symptomatic individuals due to the delay in obtainment of the second test during the home visit, we added this concern to the discussion section of the paper. (302-304) Furthermore, we emphasized the difference between the two cohorts throughout the revised manuscript.

3. What is meant with clinical sensitivity is not completely clear.

Thank you for this justified remark. With clinical sensitivity and specificity, we refer to sensitivity and specificity on clinical samples, as opposite to analytical performance. We realise this might be an unclear formulation. We deleted the terms clinical sensitivity and specificity from the manuscript and replaced them with sensitivity/specificity on clinical samples. 

4. A reference to the symptoms that were accepted/suggested to being tested in the region at the time of testing is informative and can be added in the references.

This information was added to the manuscript. (lines 109-110)

5. Data on the patient population characteristics are missing: any age distribution?

Thank you for this valuable remark. For part one of the study the included participants were men and women aged 18 years and above, we do unfortunately not have any more demographic data available. For part two of the study the participants ages varied from 18 to 84 years (M = 44, SD=16). 

The available demographic data was added to the manuscript.(line 215 and 251-252)

6. A cut off for timing was 7 days after onset of symptoms: there is no explanation why 7 days was chosen – given this categories a continuous variable into a binary variable and leads to loss of information, at minimum a reason for this choice should be made (this can be based on evidence on infectiousness or Ct evolution and discussed with your Fig 3).

The cut-off of 7 days was based on the results of Bullard et al. (Bullard, J. et al. Predicting infectious SARS-CoV-2 from diagnostic samples. Clin. Infect. Dis. 71, 2663–2666 (2020)) that showed no viral growth after incubation on Vero cells in positive samples obtained more than 8 days after symptom onset. This information was added to the manuscript. (line 206-208)

7. The specimen sample type is mentioned in the manuscript – but preferably also has a place in the abstract, mainly given the importance of specimen type in the accuracy of diagnostic tests.

This information was added to the abstract. (line 38)

8. The STARD checklist should be completed. Major comments on the elements that are part of STARD are (additional others are integrated in the detailed comments):

*Incomplete reporting of elements in the title

*Introduction: the intended use and clinical role are, as mentioned prior, mixed and it is best to improve clarity.

*Reference standard misses detail, as well participant description

* The title was adapted accordingly to ‘Evaluation of the test accuracy of a SARS-CoV-2 rapid antigentest in symptomatic community dwelling individuals in the Netherlands’ (line 1 and 2)

* For the reply to this comment we kindly refer to the response to comment one. 

* Thank you for this valid remark, we elaborated on the RT-qPCR methods used. (line 136-141)

9. It is key that this was investigated in symptomatic individuals – and this information is best included in the title. The term “community testing” is a bit misleading – given it is in a primary care setting that was serving clinical diagnosis. Mainly because already in the introduction community testing is described as a general testing strategy.

‘Symptomatic’ was added to individuals in the title (line 1 and 2) 

As explained in the response to comment one, the goal of testing at the MHS test centres is prompt isolation and initialisation of contact tracing, patients requiring medical attention are tested by the MD they attend. 

10. “Application for large- scale community testing for disease control purposes “ This is also not what this article is about – rather a place in the discussion.

We would kindly like to refer to the answer formulated on comment one. 

11. “ to qRT-PCR “: best to add if in house or a commercial assay. 

Thank you for this valid remark. Commercial assays were used, we added this information to the manuscript and elaborated further on the RT-qPCR methods used. (line 136-141)

12. What is low Ct value: best to give cut off

Thank you for this remark, the sentence was deleted from the manuscript.

13. What is magical about 7 days: is this 7 days because days since onset of symptoms was investigated as a continuous variable? (to add in text)

The cut-off of 7 days was based on the results of Bullard et al. (Bullard, J. et al. Predicting infectious SARS-CoV-2 from diagnostic samples. Clin. Infect. Dis. 71, 2663–2666 (2020)) that showed no viral growth after incubation on Vero cells in positive samples obtained more than 8 days after symptom onset. This information was added to the manuscript. (line 206-208)

14. line 65: to establish COVID-19 infection: best to add “acute” and is not COVID-19 infection, but SARS-CoV-2 infection, given COVID-19 is the disease.

Thank you for these valid remarks, the manuscript was adapted. (line 65) 

15. line 69: the PCR technology is not limited to specialized laboratories – the batched testing is, however point-of-care PCR instruments and assays are available – when receiving a CLIA waiver they can be used close to the patient. The sample can be precurred everywhere – even by self-sampling. It is the whole cascade with mostly high-throughput batched PCR and non-self sampling that do not allow for broader community-based testing and that is not focused on clinical diagnosis but on community screening in the context of a public health intervention that is referred to but not the real setting of your work.

We have discussed this remark in our group and we are unsure what the point is the reviewer puts forward here, therefore we were not able to formulate a response.

16. line 72: the word “pressurizes”: I understand the meaning, but it is not a term used as such: better to replace by “stresses or burdens”.

Thank you for this correction, this was replaced in the manuscript. (line 72)

17. line 74: rapid testing: rapid testing and reporting (if you indeed want the whole circle to be happening fast).

‘Reporting’ was added to the manuscript. (line 75)

18. line 79: what is meant with qualitative? (might be literal translation) What is probably meant and needed: high quality quantitative data

Thank you for this correction, quantitative was replaced by high quality. (lines 35, 72, 283) 

19. line 80: missing “the” in front of clinical setting 

‘The’ was added in the manuscript. (line 81)

20. line 86: “respiratory specimen” is rather broad: at least upper resp tract specimen and better to name which specimen type is used (saliva vs nose vs throat vs NPS…): This is relevant information that needs to be mentioned here in this last paragraph where the objective and PICOT of the study is stated. (it does come back in your manuscript later, which is appreciated).

Thank you for this valuable remark, this sentence was rewritten in accordance with the comments of one of the other reviewers. 

21. line 98: what is means with clinical sens for the Ct values? How is clinical sens here defined?

We have discussed this remark in our group and we are unsure what the point is the reviewer puts forward here, therefore we were not able to formulate a response.

22. line 108: missing a prior to specimen

‘ The’ was added. (line 112)

23. line 112: an additional word about: testing strategy that was in place at that moment; is this rural – city? – population catchment area? 

All community dwelling individuals with covid-19 like symptoms (cfr supra) are tested at the MHS test centres, clinically ill patients requiring medical attention are tested by the M.D. they attend. In the test centres individuals from the cities they were localised in, as well ass induvial from the more rural surroundings are tested. The total test capacity was of the tree test centres was 1200 tests per day. 

24. line 120: the manufacturer does not report a CI

The 95% CI were added to the manuscript. (line 129)

25. line 124: check English expression: visible to the naked eye

This was adapted in the manuscript. (line 133)

26. line 128: the reference standard is insufficiently described. The described platforms are not assays – a platform cannot have a target. On a platform one can use an assay – it is thus necessary to name the assay. Additional: reference to where the assay is described – approved by authorities? – some reference to its performance – mainly if validated for the specimen type and performance for the in the study used specimen collection process.

Thank you for this valuable remark, we added the used assays to the manuscript and elaborated further on the RT-qPCR methods used. (line 136-141)

27. line 151: one time “in” too many

The paragraph was rewritten. 

28. line 166-167: when there is mentioning of the sample size: why was this sample size chosen if such a specific number. To elaborate.

We would like to thank the reviewer for this valuable remark, the methods used to determine the sample size were added to the manuscript. (line 195-197)

29. line 175: “for a range of prevalence” : this is the methods section, so one can as well be precise and quote the range. As well: it is stated that agreement is assessed: using which methods? The analysis section misses the methods used to assess agreement and the calculations. More info on the methods is needed here. How are CI calculated?

‘For a range of prevalence was changed by ‘for a population prevalence of 10% and 20%’. Furthermore, we elaborated on the analytical methods used. (Line 200-211)

30. line 184: re-write - more literal translation

Thank you for this correction, the sentence was rewritten as ‘Two (0·6%) specimens with a negative VRD result were excluded because qRT-PCR could not be recovered (error in sample number registration).’ (line 217-218)

31. line 189: thank you to report on uninterpretable data – important to mention and that this proportion was very small.

32. line 209: Best to refer to: PCR pos - the pos rate by VRD was x among PCR positives.

We have discussed remarks 31 and 32 in our group and we are unsure what the point is the reviewer puts forward here, therefore we were not able to formulate a response.

33. line 224: it is unwarranted and incorrect to merge the 2 cohorts and give the PPV and NPV – which merges 2 groups with a very different pre-test probability – which, and thank you to report is unrealistically high: 60%. (latter comment on line 234 and on).

Thank you for this justified and valuable remark, the paragraphs on the combined data were removed from the manuscript. 

The reference to a population prevalence of 60% was deleted from the manuscript. 

34. line 225: plural of specimen is specimens

Thank you, this was adapted in the manuscript. (line 225)

35. line 239: here it is clear that you are assessing its performance in a high prevalence setting – thus clinical setting – because the prevalence reported on is still too high for general screening, where it will be lower than 10%. The NPV of course will be even more improved, given very low prevalence.

Thank you for this remark, we hope it is sufficiently answered with the clarification of part one and two of the study, the removal of the part on combined data and the clarification on the setting (reviewer comment one). 

36. line 264: the prospective arm here is overstated. It remains unclear if this is your target setting or not.

We would like to kindly refer to the response to comment 35.

37. line 273: “As this cut off was based…”: unclear what the message is of this sentence.

As the cut off of 30 was based on the results found in the study (data driven), a prospective evaluation is needed. 

38. line 282: A key element to mention is if the selection criteria to be tested were the same at the Breda sample in this later period – where we know that testing criteria have shifted over time. This discussion and clarity about test criteria might have a better place earlier in the discussion.

We would like to thank the reviewer for this comment. The criteria to get tested did not change during the study period, this was added to the method section. (line 111)

39. line 319: What is written supports the opportunity and the counterfactual of: with more tests, even with lower sens, this strategy will capture an absolute total of infected individuals that is larger compared to those now. It is appreciated that this is now discussed purely in the scheme of the symptomatic non-tested individuals – where this manuscript has the data to support its use in that specific group (compared to not having data to support PH community screening). This sentence might need some re-formulation however.

We have discussed this remark in our group and we are unsure what the exact point is the reviewer puts forward here, we hope the remark is sufficiently answered by the response to comment one.

40. line 326-328: is there data to back this up? Reference? At least some more elaborated discussion – reference – opinion – study – even modelling…

Thank you for this justified remark, the reference (Mina MJ, Parker R, Larremore DB. Rethinking Covid-19 Test Sensitivity - A Strategy for Containment. N Engl J Med. 2020 Sep 30) was added behind the sentence. (line 361)

41. line 330: this conclusion is not warranted that this test has – had an impact on disease control – the study did not look at this outcome in any way – but assumingly this is why it is written that it is promising – but the conclusion of your study should rather be based on what your data support. Rather as well a proposal on how further research can be meaningful and feasibly be performed and implemented can be of added value.

Thank you for this remark. As stated in the answer to comment one, the goal of testing at the GGD test centres is prompt initiation of infection control measures. As we did not look at this outcome, we write the test is promising in this context (opposite to the indication of testing of clinically ill patients with a need for medical care). The ‘impact on disease control’ was added to the topics needing further research. (line 366-367)

42. line 331: Performance: might it be that rather is meant “Implementation of the test”…

‘Performance’ was changed to ‘execution’. (line 364)

43. Table 1: this is not a table that shows a real comparison – given it purely lists the effect estimates – it is not a real agreement evaluation neither. Cross tabulation is necessary.

Thank you for this remark, cross tabulations for both visual interpretation and interpretation with VRD were added as supplementary tables. (S1)

44. Figure 1: the legend and info below a flow diagram and all tables and figures should be complete and self-explanatory. Part one better to be replaced by prospective cohort… (readers might be only looking at the graphs). As well: potentially eligible: they were or they were not – their status of being eligible was not potential: it was real. Or what is the potential of being eligible?

The captions for figure 1 and 2 were adapted, ‘prospective cohort’ and ‘qRT-PCR positive participants only’ were added respectively. 

Potentially eligible was adapted to eligible.

---

## [Decision Letter · Decision Letter 1]

29 Mar 2021

PONE-D-20-36174R1

Evaluation of the test accuracy of a SARS-CoV-2 rapid antigen test in symptomatic community dwelling individuals in the Netherlands

PLOS ONE

Dear Nathalie Van der Moeren,

Thank you for submitting your manuscript to PLOS ONE. After careful consideration, we feel that it has merit but does not fully meet PLOS ONE’s publication criteria as it currently stands. Therefore, we invite you to submit a revised version of the manuscript that addresses the points raised during the review process.

Minor comments about grammatical errors and sections with missing information have been raised by peer-reviewer #3. Please note that PLOS ONE does not copy-edit accepted manuscripts, hence we would appreciate your revisions to the highlighted areas.

We look forward to receiving your revised manuscript.

Kind regards,

Eleanor Ochodo, M.D., PhD

Academic Editor

PLOS ONE

Journal Requirements:

Reviewers' comments:

Reviewer's Responses to Questions

**Comments to the Author**

1. If the authors have adequately addressed your comments raised in a previous round of review and you feel that this manuscript is now acceptable for publication, you may indicate that here to bypass the “Comments to the Author” section, enter your conflict of interest statement in the “Confidential to Editor” section, and submit your "Accept" recommendation.

Reviewer #1: All comments have been addressed

Reviewer #3: All comments have been addressed

2. Is the manuscript technically sound, and do the data support the conclusions?

Reviewer #1: Yes

Reviewer #3: Yes

3. Has the statistical analysis been performed appropriately and rigorously? 

Reviewer #1: Yes

Reviewer #3: Yes

4. Have the authors made all data underlying the findings in their manuscript fully available?

Reviewer #1: Yes

Reviewer #3: Yes

5. Is the manuscript presented in an intelligible fashion and written in standard English?

Reviewer #1: Yes

Reviewer #3: Yes

6. Review Comments to the Author

Reviewer #1: I thank the authors for their detailed responses to my comments and improvements made to the manuscript. I have no further comments or suggestions.

Reviewer #3: Thank you for the reviewed manuscript. The decisions to exclude part 3 and other major changes improved the interpretation and quality of the manuscript and its interpretation.

The questions where answered.

Short remarks (mainly language - short words missing):

-Line 184 tracked changes version: (full participant information letter on the website, ...): will there be an url address be added?

-line 186: "No written informed consent was obtained as this would have compromised the strictly needed high flow of individuals being tested at...": the explanation is not necessary here - as long as this was discussed with the ethical committee and the written consent was waved, then this is valid. It is written later in the text as well. NOt necessary here.

-line 233: "he cut-off of 7 days was chosen based...": results OF a study by Bullard... (some re-writing - English- of the sentence necessary)

-line 288: "18 to 83 years (M = 44,": mean or median?

-line 296: "to be the ": take out the (and as well from line 301)

-line 337: "Based the cohort in part two of the study":re-write - word missing...

-line 384: "overestimate test accuracy in final clinical setting.".. grammar

7. PLOS authors have the option to publish the peer review history of their article (what does this mean?). If published, this will include your full peer review and any attached files.

Reviewer #1: **Yes: **Jac Dinnes

Reviewer #3: No

---

## [Author Response · Author response to Decision Letter 1]

2 Apr 2021

Additional points raised by Reviewer 2

-Line 184 tracked changes version: (full participant information letter on the website, ...): will there be an url address be added?

Thank you for this comment. As the participant information letter was written in Dutch, we doubt the added value of adding the URL. (https://www.google.com/url?sa=t&rct=j&q=&esrc=s&source=web&cd=&ved=2ahUKEwjnt9aWydrvAhUIg_0HHc0aAaUQFjAAegQIAxAD&url=https%3A%2F%2Fwww.ggdwestbrabant.nl%2Fcoronavirus%2F-%2Fmedia%2FA70F8404BCC24075B03934AA764D9FBD.ashx&usg=AOvVaw1Hx9cTDPy_qh-0miug_rY0)

-line 186: "No written informed consent was obtained as this would have compromised the strictly needed high flow of individuals being tested at...": the explanation is not necessary here - as long as this was discussed with the ethical committee and the written consent was waved, then this is valid. It is written later in the text as well. NOt necessary here.

Thank you for this comment. The extra information on why no written informed consent was obtained was added to the manuscript at the request of reviewer 2 and the academic editor and was clarified for study part one and two seperately. 

-line 233: "he cut-off of 7 days was chosen based...": results OF a study by Bullard... (some re-writing - English- of the sentence necessary)

Thank you for this remark, the sentence was rewritten (line 206). 

-line 288: "18 to 83 years (M = 44,": mean or median?

M stands for mean, this was added to the manuscript (line259).

-line 296: "to be the ": take out the (and as well from line 301)

Thank you for this correction, ‘the’ was deleted from both lines. (line 263 and 267) 

-line 337: "Based the cohort in part two of the study":re-write - word missing...

Thank you for this correction, the sentence was adapted. (line 286) 

-line 384: "overestimate test accuracy in final clinical setting.".. grammar

The sentence was rewritten. (line 324)

---

## [Editor Report · Decision Letter 2]

7 Apr 2021

PONE-D-20-36174R2

Evaluation of the test accuracy of a SARS-CoV-2 rapid antigen test in symptomatic community dwelling individuals in the Netherlands

PLOS ONE

Dear Nathalie Van der Moeren,

Thank you for submitting your manuscript to PLOS ONE. After careful consideration, we feel that it has merit but does not fully meet PLOS ONE’s publication criteria as it currently stands. Therefore, we invite you to submit a revised version of the manuscript that addresses the points raised during the review process.

PLOS ONE does not copy edit accepted manuscripts. We therefore request a minor modification of the statement in the ethics section lines 158-159" In part one of the study, individuals were informed about the study **trough** local media, by 159 MHS communication channels (full participant information letter on the **website, …)**. We accept not adding the URL as the letter will be in Dutch. However, the statement as it is, (website.....) may confuse readers into thinking there is missing data. Kindly delete the dots after website. In addition please change the word trough to through. A final spelling and grammar check for your manuscript will be appreciated before you resubmit.

We look forward to receiving your revised manuscript.

Kind regards,

Eleanor Ochodo, M.D., PhD

Academic Editor

PLOS ONE

---

## [Author Response · Author response to Decision Letter 2]

10 Apr 2021

- PLOS ONE does not copy edit accepted manuscripts. We therefore request a minor modification of the statement in the ethics section lines 158-159" In part one of the study, individuals were informed about the study trough local media, by 159 MHS communication channels (full participant information letter on the website, …). We accept not adding the URL as the letter will be in Dutch. However, the statement as it is, (website.....) may confuse readers into thinking there is missing data. Kindly delete the dots after website. 

The dots in question were removed. (line 159)

- In addition please change the word trough to through. 

Thank you for this correction, trough was changed by through. (line 158) 

- A final spelling and grammar check for your manuscript will be appreciated before you resubmit.

We performed a final spelling and grammar check, minor corrections were made.

---

## [Editor Report · Decision Letter 3]

16 Apr 2021

Evaluation of the test accuracy of a SARS-CoV-2 rapid antigen test in symptomatic community dwelling individuals in the Netherlands

PONE-D-20-36174R3

Dear Nathalie Van Der Moeren,

We’re pleased to inform you that your manuscript has been judged scientifically suitable for publication and will be formally accepted for publication once it meets all outstanding technical requirements.

Kind regards,

Eleanor Ochodo, M.D., PhD

Academic Editor

PLOS ONE

---

## [Editor Report · Acceptance letter]

6 May 2021

PONE-D-20-36174R3 

Evaluation of the test accuracy of a SARS-CoV-2 rapid antigen test in symptomatic community dwelling individuals in the Netherlands 

Dear Dr. Van der Moeren:

I'm pleased to inform you that your manuscript has been deemed suitable for publication in PLOS ONE. Congratulations! Your manuscript is now with our production department. 

Kind regards, 

on behalf of

Dr. Eleanor Ochodo 

Academic Editor

PLOS ONE